# The *Bordetella* type III secretion system effector BteA targets host eosinophil-epithelial signaling to promote IL-1Ra expression and persistence

Katelyn M. Parrish[1], Nicholas First[1], Jana Kamanova [2], Tyler L. Williams[1], Sarah Johnson[1], Jaylyn King[3], Karen M. Scanlon[3], Nurit P. Azouz [4,5], Seema Mattoo[6,7], Ciaran Skerry [3] & Monica C. Gestal [1] ✉

Eosinophils are traditionally associated with parasitic infections and allergic pathologies. However, emerging evidence highlights their underappreciated roles during mucosal bacterial infections. Using in vivo and in vitro approaches, we demonstrate that classical *Bordetella* spp. increase IL-1Ra production from both epithelial cells and eosinophils to facilitate immune evasion and persistence. Depletion of IL-1Ra via genetic knockout or antibody neutralization in vivo accelerated bacterial clearance. We show that the *Bordetella* type III secretion system (T3SS) effector, BteA, promotes AkT/mTOR pathway activation leading to IL-1Ra expression, which is independent of IL-1α or IL-1β production. Together, our findings uncover the molecular mechanism by which classical Bordetellae exploit host epithelial-eosinophil signaling to exclusively upregulate IL-1Ra and dampen host inflammation for persistence. These results provide therapeutic targets for controlling disease caused by long-term *Bordetella* infection and may have broader applications for other respiratory pathogens. Moreover, these insights expand our understanding of eosinophil function beyond traditional paradigms.

Respiratory infections are a leading cause of morbidity and mortality worldwide[1]. Factors such as accelerating antibiotic resistance[2], declining child vaccination rates[3,4], and waning immunity[5] heighten the potential for resurgence of respiratory pathogens[6]. Co-evolution between bacteria and their hosts creates significant selective pressure driving the development of pathogenic mechanisms that enable effective evasion of host immune responses, ensure bacterial persistence, and facilitate reinfection[7–9]. Rapid alterations in gene expression enable adaptation and survival[10], with *Bordetella* spp. being prime examples[3,5–8]. The "classical" *Bordetella* species consist of *B. pertussis*, *B. parapertussis*, and *B. bronchiseptica*, with all three having the ability to cause a characteristic chronic infection[11]. This is due to the ability of classical Bordetellae to readily fine-tune a myriad of virulence factors depending on their environment, targeting specific host cells to delay adaptive immune responses[9,12,13]. The longevity of the pneumonic stage[14,15], the subsequent residual lung pathology[16,17], and the high transmissibility[18] of *Bordetella* spp. infections support their use as a model to identify the bacterial and host factors that contribute to these disease outcomes.

*B. pertussis* is the human-specific etiological agent of whooping cough or pertussis disease[11,19–21]. *B. bronchiseptica* infects a broad range of mammals[22,23], causing "kennel cough" in canines[24,25], atrophic rhinitis in swine[26–28], and long-term pneumonic infections in mice[13]. *B. bronchiseptica* shares a common *B. bronchiseptica*-like ancestor with *B. parapertussis* and *B. pertussis*[29,30], and the genes shared among the three classical Bordetellae species have over 98% nucleotide identity[31–33]. These highly conserved regions encode virulence factors such as those that make up the type III secretion system (T3SS) machinery[32], underscoring its importance for survival and successful infection[34–36]. With both human and animal infections increasing at alarming rates despite a widely distributed and preventive vaccine being available[37–41], there is an urgent need to understand the pathogenesis of classical *Bordetella* spp.

[1]Department of Microbiology and Immunology, Louisiana State University (LSU) Health Sciences Center at Shreveport, Shreveport, LA, USA. [2]Laboratory of Infection Biology, Institute of Microbiology of the Czech Academy of Sciences, Prague, Czech Republic. [3]Department of Microbiology and Immunology, University of Maryland School of Medicine, Baltimore, MD, USA. [4]Division of Allergy and Immunology, Cincinnati Children's Hospital Medical Center, Cincinnati, OH, USA. [5]Department of Pediatrics, University of Cincinnati College of Medicine, Cincinnati, OH, USA. [6]Department of Biological Sciences, Purdue University, West Lafayette, IN, USA. [7]Department of Biochemistry, Purdue University, West Lafayette, IN, USA. ✉e-mail: monica.cartellegestal@lsuhs.edu; mcarges@gmail.com

infections. Thus, the murine *B. bronchiseptica* infection model[42] is a robust in vivo setting[42] for investigating the molecular, cellular, and biochemical mechanisms that drive immunosuppression and promote persistence[43–47].

One immunosuppressive mechanism includes the pathway regulated by the BtrS sigma factor[43,45,48,49], which includes but is not limited to the T3SS[43,50,51]. We have shown that BtrS blocks eosinophil effector functions to prevent rapid adaptive immune responses[45]. In the absence of eosinophils, RB50Δ*btrS* fails to clear due to the suppression of adaptive immune responses[45,52], suggesting that eosinophils contribute to immune responses that protect against classical Bordetellae[9]. Eosinophils are well-known for their contributions to parasite clearance and allergic pathologies[53,54], with more recent evidence supporting their roles in preserving mucosal homeostasis and tolerance[55]. However, their ability to suppress inflammatory responses also suggests that eosinophils may be ideal targets of pathogens, as seen during *Clostridioides difficile* infections[56].

Eosinophils are strongly associated with a multitude of airway pathologies, including asthma[57] and chronic obstructive pulmonary disease[58]. Moreover, dysregulation of interleukin 1 (IL-1) signaling is a shared hallmark of severity for these diseases[59]. During *Bordetella* spp. infection, IL-1 signaling[60] contributes to two polarizing roles in pathology and clearance. IL-1 signaling encompasses three major cytokines, IL-1α, IL-1β, and IL-1 receptor antagonist (IL-1Ra)[61]. All three cytokines compete for binding to the IL-1 receptor 1 (IL-1R1) to initiate the subsequent signaling cascade. Binding of IL-1α or IL-1β to IL-1R1 initiates a cascade of pro-inflammatory signal responses, whereas IL-1Ra competes with IL-1α and IL-1β and blocks downstream IL-1R-mediated inflammatory signaling[60]. During *Bordetella* spp. infection, IL-1α does not contribute to clearance[62], but it does somewhat contribute to the remaining epithelial lung pathology found after infection. IL-1α-derived pathology is caused by the *Bordetella* spp. tracheal cytotoxic toxin (TCT)[63] via nucleotide-binding oligomerization domain-containing protein 1 (NOD1) activation[64]. While it has been suggested that TCT causes lung pathology, new evidence has also shown that pathology is due to NOD1 activation and independent of TCT activity. However, the specific factors contributing to pathology via IL-1α remain largely unknown. Conversely, IL-1β is required for triggering neutrophil-associated responses and vaccine-mediated immune protection[65], as well as natural killer cell activation[66]. This implies that IL-1β has a different effect, and it is mostly associated with clearance and protective responses to *Bordetella* spp. infections. Importantly, previous literature points toward epithelial cells as major sources of IL-1α[63], while macrophages appear to be the drivers of IL-1β-mediated immune response[65,66]. These studies point toward specific *Bordetella* spp. factors that are capable of inducing unique IL-1-associated responses, solely based on the cell type being targeted.

Previous literature has indicated that eosinophils, as well as epithelial cells, can drive IL-1Ra responses[57,67–69]. Based on the role of eosinophils in inducing persistence of the wild-type infection[9], we investigated the role of lung eosinophil-epithelial cell signaling during murine infection with *B. bronchiseptica* strain RB50. Combining multidisciplinary approaches, this work unravels the bacteria- and host-specific mechanisms by which classical *Bordetella* spp. trigger interleukin 1 receptor antagonist (IL-1Ra) production, at both the cellular and molecular levels. Here, we identify that the *Bordetella* T3SS effector, BteA, targets eosinophils and epithelial cells to increase IL-1Ra production and create an immunosuppressive environment that facilitates persistence, a strategy that is conserved amongst classical Bordetellae.

## Results

### *B. bronchiseptica* promotes IL-1Ra induction following in vivo infection

One of the main features of murine infection with the wild-type *B. bronchiseptica* RB50 strain is lung persistence. Following challenge with an RB50 mutant lacking BtrS (RB50Δ*btrS*, or "Δ*btrS*"), infection is rapidly cleared from the lungs compared to wild type (Fig. 1A), revealing an immunosuppressive function during RB50 infection, which allows for persistence. To further characterize the function of BtrS, we investigated its

specific contribution to dampening host immune responses. We have previously shown that both BtrS and eosinophils are required for long-term infection in mice[9], proposing that RB50 directly suppresses eosinophil effector functions via BtrS[57,68,70,71]. Our previous work also suggests a role for eosinophils in the modulation of IL-1-mediated responses[52,72–76]. IL-1Ra levels increase in response to bacterial and fungal infections[77–79] and even following *B. pertussis* infections[80]. Increased IL-1Ra levels have also been strongly associated with higher bacterial burden[81,82] and worsened infection outcome[76,83]. IL-1 signaling has recently been shown to be dysregulated during *M. tuberculosis* infection[84] via IL-1Ra production to promote persistence, while depletion of IL-1Ra facilitates clearance[85]. Thus, we hypothesized that *Bordetella* spp. induces IL-1Ra production through a BtrS-regulated mechanism to dampen inflammatory responses and facilitate persistence[72–76]. Using an in vivo infection time course, we sought to determine the time point by which RB50Δ*btrS* gets rapidly cleared from the lungs, while RB50 successfully persists. To do this, we infected BALB/c mice with RB50 or RB50Δ*btrS* until 28 days post-infection (dpi). While there is little difference in lung CFU at 7 dpi, RB50Δ*btrS* is completely cleared by 14 dpi, with $10^5$ CFU/mL remaining in RB50-infected lungs even by 28 dpi (Fig. 1A). Given these results, we first hypothesized that wild-type RB50 infection induces IL-1Ra expression, which allows for persistence within the lungs. To test this, we evaluated changes in IL-1Ra expression at different time points post-infection with RB50 or RB50Δ*btrS* in BALB/c mice. Following RB50 infection, transcript levels increased as early as 1 dpi, peaking at 7 dpi and returning to basal levels by 14 dpi (Fig. 1B). No significant increase was observed following infection with RB50Δ*btrS* at any time point. In order to confirm changes at the protein level, an IL-1Ra ELISA was performed using lungs from infected mouse lungs at different time points infection (Fig. 1C, Supplementary Fig. 1A). Our results also revealed that at day 7 there are no differences in IL-1α secretion in the lungs (Fig. 1D) between groups, while an increase was observed in IL-1β (Fig. 1E) following RB50 infection. This increase was expected, as we have previously shown that IL-1β induction is promoted by a *btrS*-regulated mechanism[38]. It is worth noting that IL-1α and IL-1β are pre-synthesized and stored in cells[86,87], but they require cleavage for activation. Conversely, IL-1Ra is pre-synthesized and stored in its functional form[88], allowing for the investigation of its transcripts, which accurately represent protein levels. Thus, we conclude that RB50 promotes an early induction of IL-1Ra and IL-1β following infection via a BtrS-regulated mechanism.

### IL-1Ra secretion is driven by epithelial cells and eosinophils

Given that eosinophils and epithelial cells are well-appreciated modulators of IL-1 signaling from our work[45,52] as well as others[57,68], in combination with the observed increase in IL-1Ra and its known role[70,71], we decided to investigate which cell types produce IL-1Ra following infection, as these cells could be responsible for the immunosuppression that allows for the establishment of infection. It has been shown that epithelial cells and eosinophils are the major drivers of IL-1Ra secretion[69], possibly due to their critical roles in mediating tissue healing and repair[55,88,89], and this function strongly depends on their tightly regulated communication[68]. In fact, when investigating the crosstalk between eosinophils and epithelial cells in vitro, IL-1Ra is enhanced during co-culture[68]. Altogether, this led us to hypothesize that epithelial cells and eosinophils work in coordination to drive IL-1Ra secretion during classical *Bordetella* infection.

To confirm the contribution of epithelial cells and eosinophils in IL-1Ra secretion, immunofluorescence staining for IL-1Ra, epithelial cells, and eosinophils was performed at days 1, 3, and 7 post-infection with RB50 or RB50Δ*btrS*. Increased IL-1Ra signal from epithelial cells and eosinophils started as early as 1 dpi in RB50-infected mice, while those infected with RB50Δ*btrS* showed no increase (Fig. 2A), with the quantification of the microscopy images supporting our mRNA and ELISA results (Fig. 2B). We also observed that epithelial cells and eosinophils account for at least 50% of the total IL-1Ra across the time course of infection (Fig. 2A, B).

To further investigate the contributions of epithelial cells and eosinophils to the induction of IL-1Ra, we used A549 human lung-derived

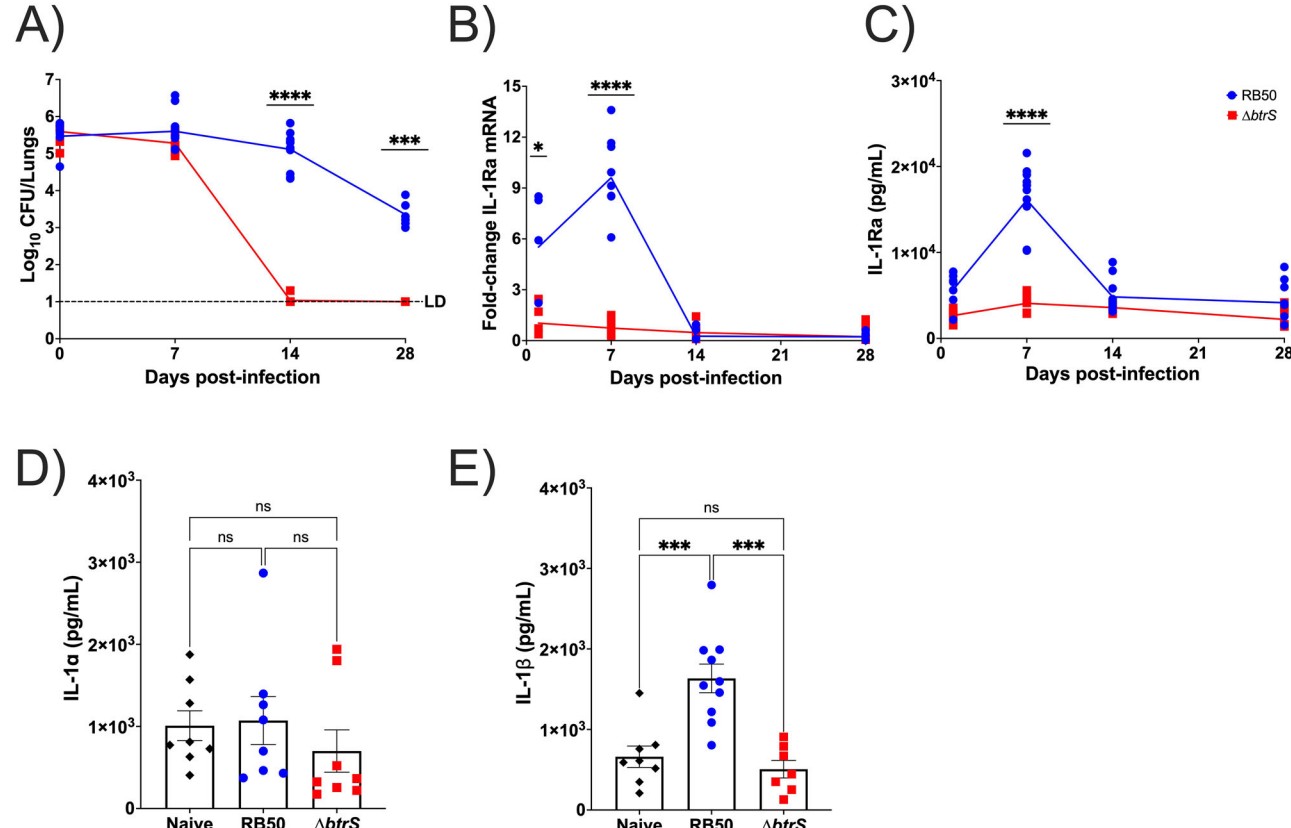

**Fig. 1 | RB50 promotes IL-1Ra expression following in vivo infection.** BALB/c mice were intranasally challenged with RB50 (blue circles) or RB50Δ*btrS* (red squares). **A** Lungs were extracted and CFU/mL was enumerated at 0, 7, 14, and 28 dpi (*n* = 6–12). **B** Lung RNA was extracted at days 7, 14, and 28 post-infection for qRT-PCR to quantify IL-1Ra expression, normalized to actin controls, and compared to uninfected groups (*n* = 8–12). **C** Lung homogenate serum was extracted for IL-1Ra ELISA at days 7, 14, and 28 post-infection (*n* = 5–10). Lung serum was collected at 7 dpi for **D** IL-1α ELISA, or **E** IL-1β ELISA (*n* = 6–10). Dunnett's One-Way ANOVA with multiple comparisons test was used for statistical analysis of all data presented in this figure panel. \**p* ≤ 0.05, \*\**p* ≤ 0.01, \*\*\**p* ≤ 0.001, and \*\*\*\**p* ≤ 0.0001.

epithelial cells and murine bone marrow-derived eosinophils (bmEos). These cells were used in monoculture and inoculated with cell culture media only, RB50, or RB50Δ*btrS*. We selected to perform our studies at 2 hours post-infection (hpi), when live bacteria and cell counts are both at comparable numbers at an MOI of 10. Our results revealed that at 2 hpi, both epithelial cells (Fig. 2C) and eosinophils (Fig. 2D) have increased mRNA levels of IL-1Ra following infection with RB50. In contrast to RB50 infection, challenge of epithelial cells or eosinophils with RB50Δ*btrS* showed no increase in IL-1Ra expression, comparable to the naïve untreated cells. Overall, our in vivo and in vitro data, revealed that following infection with RB50, epithelial cells and eosinophils induce IL-1Ra at mRNA and protein levels, suggesting that IL-1Ra induction might be exploited by *B. bronchiseptica* to promote persistence.

### IL-1Ra induction facilitates RB50 persistence

With increased levels of IL-1Ra, at early times post-infection with the wild-type *B. bronchiseptica* RB50, considering that RB50 burden remains heightened in the lungs for 56 days in mice, and combined with previous literature associating IL-1Ra with bacterial persistence[83], we hypothesized that IL-1Ra production is required for the characteristic classical *Bordetella* spp. long-term lung infection. To determine the effects of IL-1Ra on infection longevity with RB50 and RB50Δ*btrS*, we used an IL-1Ra-knockout mouse model (*Il1rn*[−/−])[81,82] for evaluating lung colonization compared to the wild-type C57BL/6J mice. We used RB50Δ*btrS* infection as a negative control in C57BL/6J mice. The 14-day time point was chosen since this is when RB50-infected mice are heavily colonized in the lungs, while those infected with RB50Δ*btrS* strain (negative control for IL-1Ra induction)[9,43] are nearly cleared (Fig. 1A and Supplementary Fig. 2B). In *Il1rn*[−/−] mice,

RB50 burden showed a 3-log decrease in the lungs compared to wild-type C57BL/6J-infected mice, reaching similar levels of colonization to wild-type mice challenged with RB50Δ*btrS* (Fig. 3A). This suggests that IL-1Ra may contribute to increased lung persistence following infection.

Based on these results, suggesting a role for IL-1Ra in persistence and previous literature[90], we tested if IL-1Ra supplementation following infection with RB50 and RB50Δ*btrS* would augment lung persistence. Daily intraperitoneal injections of IL-1Ra were given from 1 to 14 dpi, and lung colony-forming units (CFU) were enumerated. Our results showed that IL-1Ra supplementation led to an increased bacterial burden following infection with both RB50 and RB50Δ*btrS* (Fig. 3B), with a 2-log increase in RB50Δ*btrS* lung burden between untreated and treated mice. Taken together, these results, from two complementary approaches, support that IL-1Ra production is required for promoting RB50 persistence in the lungs, and the observed increase in bacterial burden following infection is *btrS*-dependent.

### IL-1Ra induction is a conserved strategy amongst classical Bordetellae

Based on our results and previous literature[77–79], we hypothesized that classical *Bordetella* spp. induce IL-1Ra as a conserved strategy to promote the characteristic long-term infection. To test the influence of different infection time points on the therapeutic effects of anti-IL-1Ra treatment, we started daily intranasal monoclonal antibody (mAb) treatment following RB50 challenge at either 1, 3, or 5 dpi with RB50 until 14 dpi for lung CFU enumeration (Supplementary Fig. 1B). The results revealed that treatment with mAb anti-IL-1Ra antibodies resulted in a significant decrease in bacterial burden in the lungs at day 14 post-infection, regardless of the start time

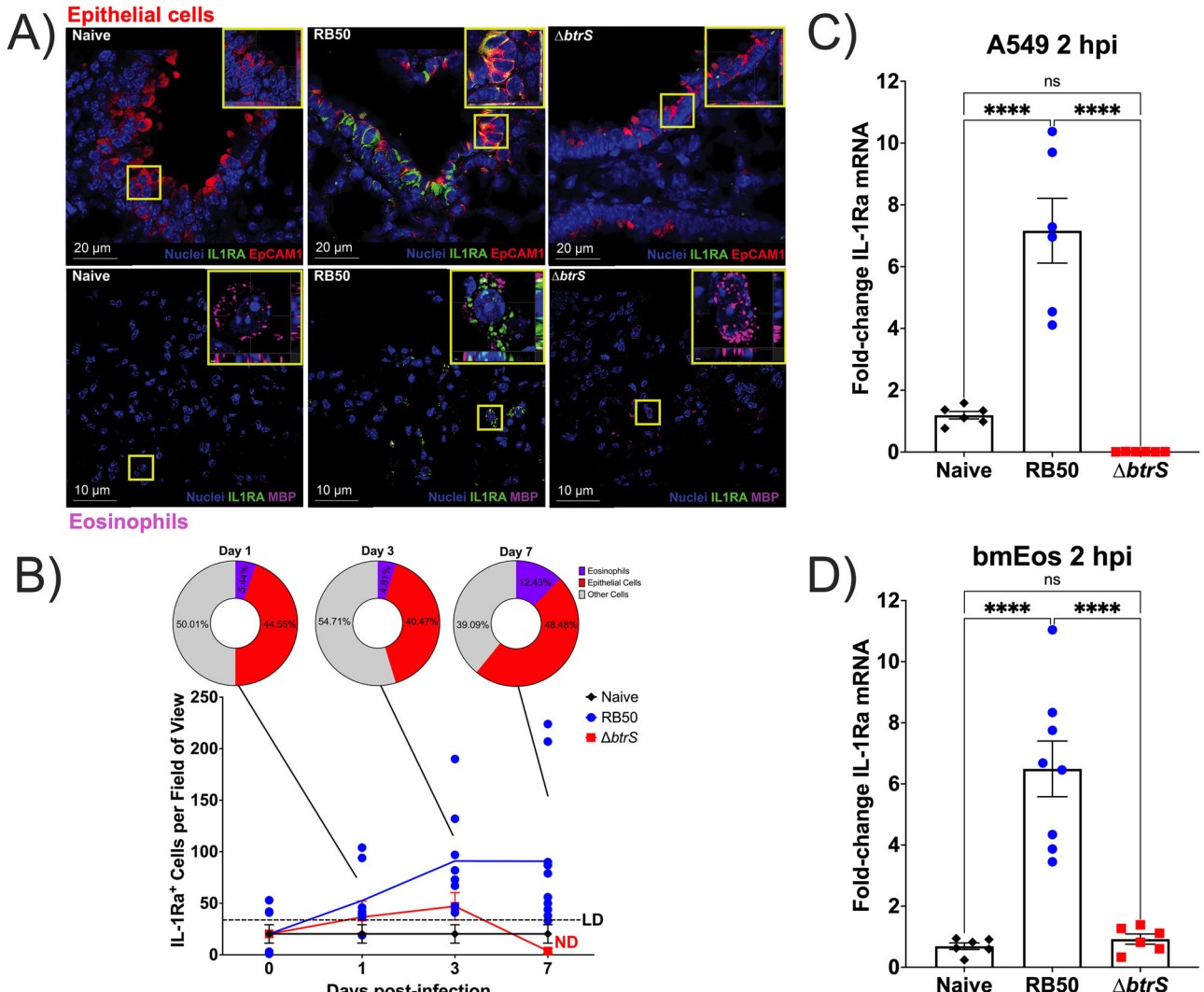

**Fig. 2 | IL-1Ra production is driven by epithelial cells and eosinophils.** BALB/c mice were intranasally challenged with RB50 (blue circles) or RB50Δ*btrS* (red squares). **A** Representative immunofluorescence microscopy images of perfused, fixed, and paraffin-embedded lungs at 7 dpi acquired with the Olympus spinning disk (*n* = 4). Naïve and infected lungs were stained for anti-IL-1Ra (green), Hoechst for cell nuclei (blue), and anti-EpCAM1 for epithelial cells (red) or anti-MBP for eosinophils (purple). **B** Acquired images of RB50-infected (blue circles) and Δ*btrS*-infected (red squares) mice were quantified using Imaris software, with colocalization criteria determined as IL-1Ra+ signal within 0.5 μm from the cell nucleus. Each symbol represents 10-15 areas captured using 3 technical replicates (*n* = 4 mice). The pie charts depict the proportion of eosinophils (purple), epithelial cells (red), and undetermined cells (gray) to total IL-1Ra-positive cells (100%, *n* = 8–10). **C** A549 cells or **D** bone marrow-derived eosinophils (bmEos) were challenged with RB50 (blue circles) or RB50Δ*btrS* (red squares) at an MOI = 10. At 2 hpi, RNA was extracted for qRT-PCR. mRNA levels of IL-1Ra were normalized to actin, shown as fold-change IL-1Ra gene expression compared to uninfected actin controls (black diamonds). Each symbol represents the average of three biological replicates run in duplicate (*n* = 6–8). Each bar represents the mean ± SEM. One-Way ANOVA with Tukey's multiple comparison test was performed for (**C**, **D**). ns non-significant, ****$p < 0.0001$.

of treatment following infection. Given these results, we decided to start mAb antibody treatment at day 5 post-challenge, based on the literature[91] and we increased the volume to 30 μL to ensure homogeneous lung distribution[92] during our experiments.

As a control of the anti-IL-1Ra mAb treatment, BALB/c mice were intranasally infected with the RB50 strain and treated intranasally with 30 μL mAb anti-IL-1Ra (100 ng)[83,93] or 30 μL of an IgG2a isotype control from 5 to 14 dpi (Fig. 3C). Our results reveal that treatment with mAb anti-IL-1Ra antibodies significantly reduced bacterial burden following infection with RB50. Mice treated with the isotype control showed no change in clearance. To determine if induction of IL-1Ra is a common strategy utilized by classical *Bordetella* spp., we treated with anti-IL-1Ra following infection with RB50 or a streptomycin-resistant, Tohama I-derived *B. pertussis* strain Bp536[94]. Mice infected with Bp536 showed similar results to RB50 in the untreated and infected controls (Fig. 3D).

Given that laboratory strains promote induction of IL-1Ra, we wanted to determine if circulating isolates of *B. pertussis* also promote IL-1Ra induction and if the use of intranasal anti-IL-1Ra antibodies could decrease bacterial burden in the lungs. Three clinical isolates donated by the CDC were randomly selected to maintain an unbiased approach for our experiments (Table 1). Of the three selected, BpH787 is a *prn*-positive, *fim2*-serotype, *ptxP3*-containing strain[95]; J021 is *prn*-negative (due to promoter disruption), *fim1*-serotype, *ptxP3*-containing strain[95]; and I420 is a *prn*-positive, *fim1*-serotype, *ptxP3*-containing strain[35]. Using the same experimental approach as previously explained, our results revealed that anti-IL-1Ra mAb treatment facilitated clearance of all three isolates from the lungs by 14 dpi (Fig. 3E), implicating that even circulating *B. pertussis* also exploits IL-1Ra production in addition to lab-adapted *B. bronchiseptica* and *B. pertussis*. Altogether, our results indicate that classical *Bordetella* spp. possess a conserved and maintained

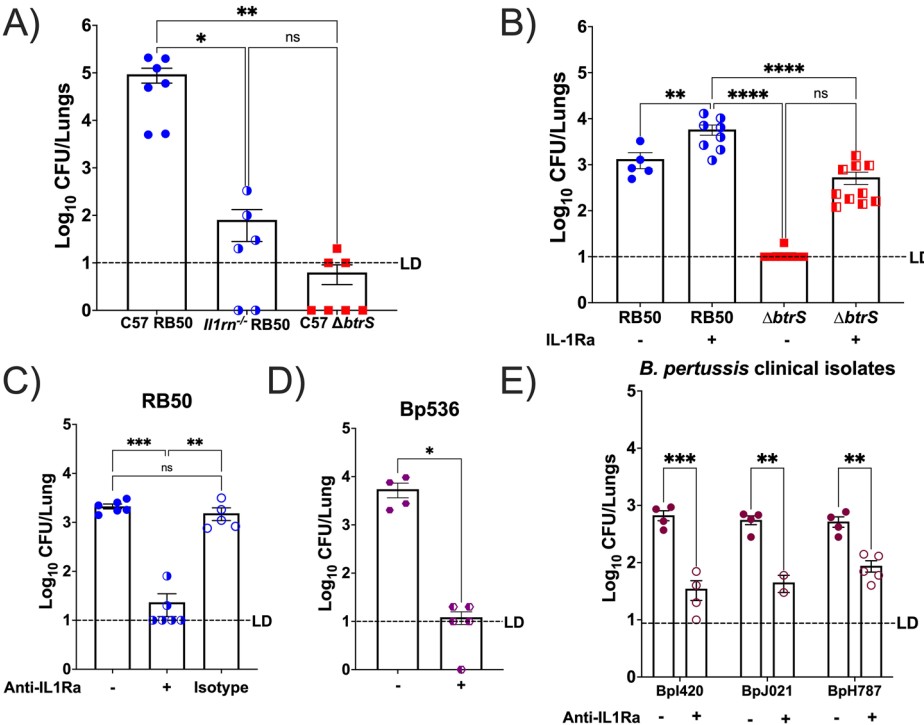

**Fig. 3 | IL-1Ra induction is conserved amongst classical Bordetellae. A** C57BL/6 J (filled) or $Il1rn^{-/-}$ (half-shaded) mice were intranasally challenged with RB50 (blue circles) or RB50ΔbtrS (btrS) (red squares). At 14 dpi, lung CFU were enumerated. ($n = 6–7$). **B** BALB/c mice were challenged with RB50 (blue circles) or RB50Δ*btrS* (red squares), either infected alone (filled) or supplemented with daily intraperitoneal IL-1Ra (half-shaded) from 1 to 14 dpi. Lung CFU were enumerated at 14 dpi ($n = 5–10$). BALB/c mice were intranasally challenged with **C** *B. bronchiseptica* RB50 or **D** *B. pertussis* 536, or with **E** three clinical CDC *B. pertussis* isolates. 30 μL of 500 μg anti-IL-1Ra was intranasally administered to treated groups from 5 to 14 dpi, and lung CFU were enumerated at 14 dpi. Each symbol represents the average of three biological replicates run in duplicate ($n = 4–5$), and each bar represents the mean ± SEM. Tukey's One-Way ANOVA was used for statistical analysis. *$p < 0.05$, ***$p < 0.001$.

## Table 1 | Bacterial strains used in this study

| Strain | Species | Phenotype | References |
|---|---|---|---|
| RB50 | *B. bronchiseptica* | Wild type | 8,44 |
| RB50Δ*btrS* | *B. bronchiseptica* | *btrS* knockout | 43 |
| RB50Δ*bscN* | *B. bronchiseptica* | *bscN* knockout | 108 |
| RB50Δ*bteA* | *B. bronchiseptica* | *bteA* knockout | 113 |
| RB50Δ*bteA::bteA* | *B. bronchiseptica* | BteA complementation | This study |
| Bp536 | *B. pertussis* | Wild type (Strep$^R$ Tohama I derivative) | 8,44,97 |
| BpH878 | *B. pertussis* | Wild type (Clinical isolate) | 95 |
| BpI420 | *B. pertussis* | Wild type (Clinical isolate) | 35 |
| BpJ021 | *B. pertussis* | Wild type (Clinical isolate) | 95 |

BtrS-regulated mechanism to promote early IL-1Ra induction, which facilitates persistence in the lower respiratory tract.

## IL-1Ra induction is increased in a model of severe infant pertussis

Given that classical Bordetellae promote IL-1Ra induction, and that neonatal infection associates with high bacterial burden and enhanced mortality [96]; we investigated if IL-1Ra induction occurs in neonatal mice to facilitate uncontrolled growth of *B. pertussis* in the lungs, as previously observed in humans and the murine model[97]. At day 3 post-infection, in mice inoculated at post-natal day 7 with *B. pertussis*, RNA sequencing was performed. We used juvenile mice as controls[98]. We found that levels of IL-1Ra expression increases rapidly following infection with *B. pertussis* starting at 3 days post-infection, mimicking the early induction observed in the BALB/c infection model (Supplementary Fig. 1C). Interestingly, the data also revealed that neonate mice have higher basal levels of IL-1Ra even uninfected, which correlates with previous human studies showing increased IL-1Ra basal levels in neonates[99–102]. To validate the neonate data, we evaluated mRNA levels of IL-1Ra at days 4 and 8 post-infection with *B. pertussis*. These days were selected based on the neonate's life expectancy

following infection, as most of them succumb at 9 days post-infection[103]. Our results showed that neonates present a significant increase in IL-1Ra mRNA transcripts following infection with *B. pertussis* at both 4 and 8 dpi compared to uninfected controls (Fig. 4A), coupled with a maintained high lung bacterial burden also at both 4 and 8 dpi (Fig. 4B), suggesting that the sustained high CFU mimics the uncontrolled infection and high lung bacterial burden observed during severe cases of neonatal *B. pertussis* in humans. Immunofluorescence microscopy was performed to further confirm the contribution of epithelial cells (Fig. 4C) and eosinophils (Fig. 4D) to IL-1Ra levels in the lungs (Fig. 4C, D). The results revealed that neonates, as well as 4-6 weeks old mice, present increased signal of IL-1Ra that colocalized mostly with epithelial cells (EpCAM) and eosinophils (MBP), further supporting the previous data. Overall, our results, using alternative approaches, indicate that *B. pertussis* infection, in a secondary murine model (C57BL/6 J) of severe pertussis, also promotes IL-1Ra induction.

## The *Bordetella* T3SS effector, BteA, is required for IL-1Ra expression

Our work shows that *B. bronchiseptica* promotes IL-1Ra and IL-1β production in a *btrS*-dependent manner. However, the underlying precise molecular bacterial mechanism has yet to be identified. One of the gene clusters that is regulated by the BtrS sigma factor is the T3SS[50,51,104]. The T3SS is known for promoting anti-inflammatory responses[36,105] and in other bacterial species, it is involved in the modulation of IL-1-related cytokine secretion[78], including IL-1Ra. In support of this, our lab has shown that T3SS promotes anti-inflammatory responses that lead to increased persistence mediated by the disruption of the VIP/VPAC2 anti-inflammatory axis[104]. Interestingly, VIP modulates IL-1Ra levels, and an increase in VIP concentration leads to an increase in IL-1Ra concentration[106]. Based on the anti-inflammatory role of the T3SS[36,104] and its regulation by BtrS[50,51], we hypothesized that *B. bronchiseptica* uses its T3SS machinery to enhance IL-1Ra expression. To test this hypothesis, mice were challenged with RB50Δ*bscN*[107], an RB50 mutant lacking the T3SS ATPase required for translocation of its effector proteins into target host cells[108]. At 7 dpi, we compared fold-change lung IL-1Ra transcript levels of RB50Δ*bscN* with those of wild-type RB50 and RB50Δ*btrS* (Supplementary Fig. 2A). Mice infected with RB50 showed a drastic increase in IL-1Ra expression compared to those in the uninfected control group, while the RB50Δ*bscN* and

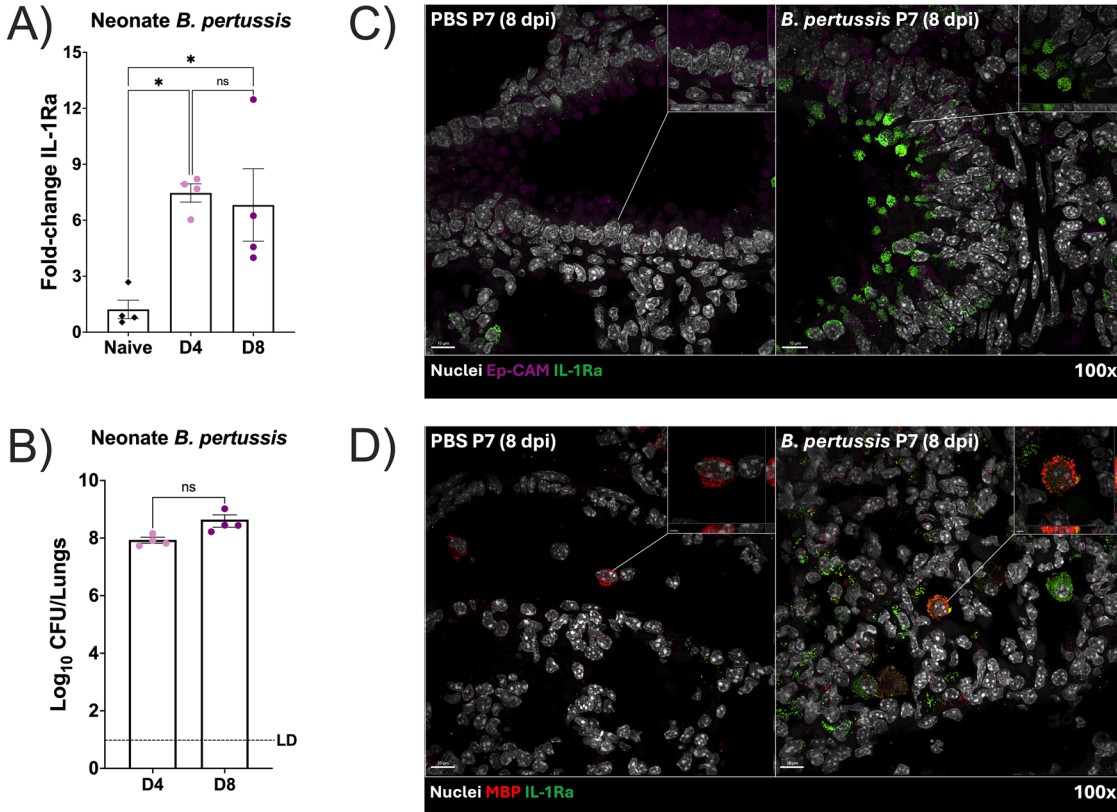

**Fig. 4 | IL-1Ra is increased in neonatal mice following *B. pertussis* infection.** C57BL/6J neonate or infant mice were inoculated with PBS (gray bars) or 6 ×10⁹ CFU/mL of *B. pertussis* 536 via aerosol nebulization. **A** At 4 dpi (light pink) and 8 dpi (dark pink), lung RNA was extracted to measure fold-change IL-1Ra expression via qRT-PCR normalized to actin controls. Each individual point represents fold-change IL-1Ra expression compared to average uninfected controls (black diamonds) from a single animal, with columns representing mean ± SEM (*n* = 4). **C** Lung CFU were enumerated following 4 dpi (light pink) and 8 dpi (dark pink).

Each symbol represents the CFU/mL of one mouse. Tukey's One-Way ANOVA with multiple comparisons test was performed for (**B**) and Wilcoxon's *t*-test was used for was performed for (**C**). ns = non-significant, *$p < 0.05$. **C, D** Images were acquired with the Olympus spinning disk. Neonate C57BL/6J (P7) mice were inoculated with PBS only or Bp536, and lungs were perfused, fixed, and embedded at 8 dpi for IF microscopy. Hoechst was used for cell nuclei (white), and samples were stained against IL-1Ra (green) and either anti-EpCAM (purple) for epithelial cells or anti-MBP (red) for eosinophils.

RB50Δ*btrS* mutant infections showed relatively no change, suggesting that increase of IL-1Ra is mediated by the T3SS.

The T3SS of classical Bordetellae, up to this date, is known to inject two protein effectors into the host cell cytoplasm: BteA[109–112] and "gatekeeper" protein, BopN[113–115], which facilitates BteA injection into the host cell[105,113,114]. Both effectors have been shown to contribute to immunomodulation[115–117]. However, the exact molecular and cellular mechanisms involved remain unclear. To investigate the injected effector responsible for the induction of IL-1Ra, we compared fold-change mRNA expression in the lungs of mice following infection with RB50 or a *bteA*-null knockout mutant[109,111,117,118], using RB50 as a positive control, and RB50Δ*btrS* negative control. Our results indicate that while the RB50 strain increases mRNA levels of IL-1Ra, all knockout strains failed to induce IL-1Ra expression (Fig. 5A), with similar results observed at the protein level (Fig. 5B). Complementation of BteA restored IL-1Ra induction. The higher levels of IL-1Ra observed in infections with the complemented strain, compared to the wild-type strain, may be due to variations in the amounts of injected BteA. We show that BteA has no significant effect on IL-1α (Fig. 5C) or IL-1β (Fig. 5D) protein levels. Compared to the decrease in IL-1β levels following RB50Δ*btrS* infection (Fig. 1E), this is not observed following infection with RB50Δ*bteA* (Fig. 5D) relative to RB50. Overall, this suggests that changes in IL-1α, IL-1β, and IL-1Ra levels in the lungs following wild-type infection are all induced by virulence factors that occur separately from one another by targeting specific cell types.

To confirm that BteA is required and sufficient to induce IL-1Ra and facilitate wild-type infection persistence, we treated mice infected with either

RB50 or RB50Δ*bteA* followed by IL-1Ra treatment with anti-IL-1Ra mAb from 5 to 14 dpi. At day 14 dpi, lung bacterial burden was compared between wild-type and mutant infection groups. We expected that if BteA is the sole driver of this response, we should see no difference in the CFUs between treated and untreated groups with no *bteA*. Indeed, we observed no differences regardless of antibody treatment with RB50Δ*bteA* compared to wild-type infection, where lung bacterial burden was significantly lower following anti-IL-1Ra administration compared to untreated, infected control mice (Supplementary Fig. 3A). To further evaluate the contribution of BteA in a minimalistic setting, we used the in vitro system challenging lung epithelial cells or bone marrow-derived eosinophils to confirm the contribution of BteA to IL-1Ra induction, and our results further confirm that BteA promotes IL-1Ra expression in epithelial cells (Fig. 5E) as well as eosinophils (Fig. 5F). Altogether, these results indicate that BteA is required for IL-1Ra induction in epithelial cells and eosinophils.

**BteA hijacks Akt signaling to induce IL-1Ra production**

It has been shown that the Akt/PI3K axis can induce IL-1Ra signaling[119]. Thus, we hypothesized that the *Bordetella* spp. effector of the T3SS, BteA, activates Akt leading to IL-1Ra induction. To test this hypothesis, we evaluated the role of Akt phosphorylation on IL-1Ra induction following infection of A549 lung epithelial cells. Akt can be phosphorylated in two different positions, threonine 308 (T308) and serine 473 (S473)[120,121]. In our work, we evaluated phosphorylation of S473, which is well-established as a marker for Akt activation[122–124]. In order to detect small changes in S473 phosphorylation levels in vitro that may be missed using qualitative assays,

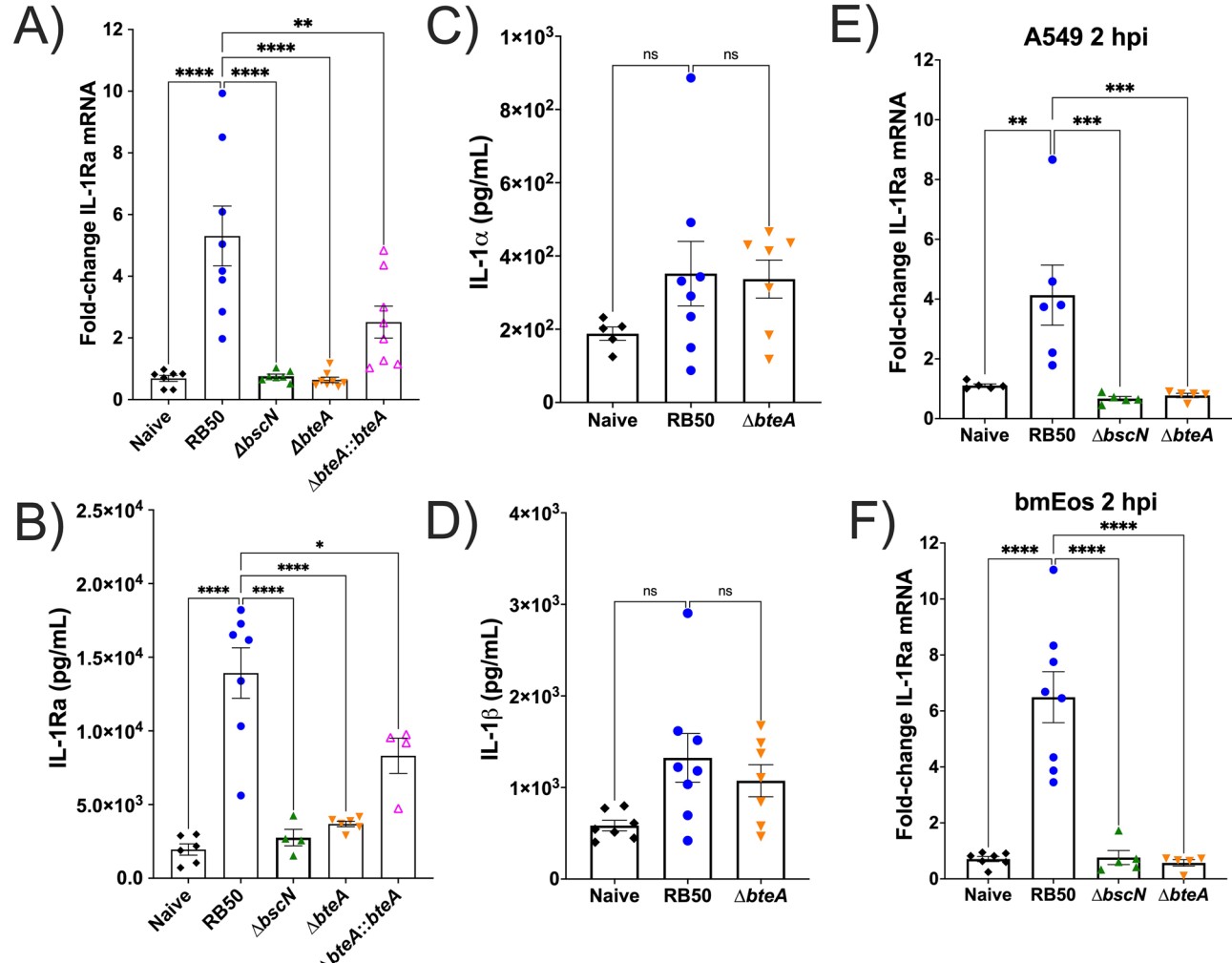

**Fig. 5 | The *Bordetella* T3SS is required for IL-1Ra expression.** BALB/c mice were intranasally inoculated with PBS alone, or 30 µL containing $10^6$ CFU/mL of RB50 (blue circles), RB50Δ*bscN* (green triangles), RB50Δ*bteA* (orange inverted triangles), or RB50Δ*bteA::bteA* (pink empty triangles). At 7 dpi, lung RNA was extracted for IL-1Ra qRT-PCR (**A**) or IL-1Ra ELISA (**B**). Each symbol represents fold-change IL-1Ra expression calculated from a single mouse ran in duplicate, with columns representing mean ± SEM ($n = 4–8$). Tukey's One-Way ANOVA with multiple comparisons test was used for (**A**, **B**). **C** Lung IL-1α or (**D**) IL-1β levels were measured at 7 dpi in BALB/c mice following intranasal inoculation with PBS (black diamonds), 30 µL of $10^6$ RB50 (blue circles), or 30 µL of $10^6$ RB50Δ*bteA* (orange inverted triangles). Each symbol represents the average of a single mouse performed in duplicate ($n = 4–8$). A549 cells (**E**) or bmEos (**F**) were incubated with media only (black diamonds), or with RB50 (blue circles), RB50Δ*bscN* (green triangles), or RB50Δ*bteA* (orange inverted triangles) at an MOI = 10. At 2 hpi, RNA was extracted for IL-1Ra expression via qRT-PCR. Each symbol represents the average of one sample run in duplicates, with at least 3 independent experiments performed for each setting ($n = 5–8$). For ELISA, each symbol represents the average absorbance of one biological replicate run in duplicates, with at least 3 individual assays performed for each protein target. ns non-significant, *$p < 0.05$, **$p < 0.01$, ***$p < 0.001$, ****$p < 0.0001$.

such as immunoblotting, we utilized an ELISA (Invitrogen, Cat. #KHO0111) to measure Akt S473 phosphorylation, following infection of A549 cells with RB50 or RB50Δ*bteA* at an MOI of 10. Our results show that at 2 hpi, Akt S473 phosphorylation was increased in a BteA-dependent manner compared to cells infected with the *bteA*-null mutant (Fig. 6A).

To confirm the contribution of Akt phosphorylation and the role of BteA, we decided to use an alternative approach using the allosteric pan-Akt inhibitor, MK-2206 (10 µM). Uninfected A549 cells were used as baseline control, and compared to RB50-infected cells, and cells infected with RB50Δ*bteA*. One group was untreated, and the other group was treated with an Akt inhibitor to subsequently evaluate levels of expression of IL-1Ra following suppression of Akt phosphorylation. Our outcome was to evaluate mRNA levels of IL-1Ra. Our results show that RB50 induces mRNA IL-1Ra expression, a mechanism that is mediated by Akt activation, as shown using the addition of an Akt inhibitor (MK-2206, 10 µM), which significantly reduced this induction (Fig. 6B). These results, together with our previous data, support that BteA promotes IL-1Ra induction via Akt phosphorylation.

## Akt signals via mTOR phosphorylation to induce IL-1Ra

Finally, we wanted to further define the signaling pathway responsible for the IL-1Ra induction. At 2 hpi, we evaluated mTOR phosphorylation signal using immunofluorescence staining and microscopy imaging. Our results revealed that RB50-infected A549 cells have increased mTOR phosphorylation signal (Fig. 7A) compared to uninfected controls and RB50Δ*bteA* infection. This is supported by quantification of phospho-mTOR signal (Supplementary Fig. 3C), with modest restoration of mTOR phosphorylation observed following *bteA* complementation.

To investigate if our observed phosphorylation of Akt is associated with mTOR phosphorylation, we utilized an Akt inhibitor (MK-2206, 10 µM) and performed a phospho-mTOR ELISA (Antibodies.com, Cat. #A102225) at 2 hpi to evaluate levels of mTOR phosphorylation at S4281. Akt inhibitor treatment resulted in decreased mTOR S4281 phosphorylation in RB50-infected cells compared to the untreated A549 infection (Fig. 7B). There were no differences between treated and untreated groups infected with RB50Δ*bteA*, suggesting this is a BteA-dependent mechanism. To correlate

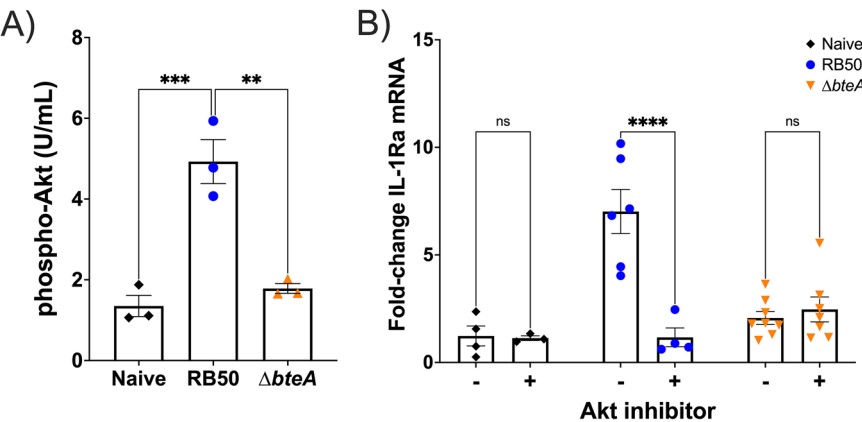

**Fig. 6 | BteA induces Akt phosphorylation for IL-1Ra production.** A549 cells were incubated with only cell culture media (black diamonds), or at an MOI = 10 with RB50 (blue circles) or RB50Δ*bteA* (orange inverted triangles). **A** At 2 hpi, cell lysates were collected to perform an ELISA to detect phosphorylated Akt (Ser473). Tukey's Two-Way ANOVA with multiple comparisons test was performed for statistical analysis. **B** One group was untreated, and the other was treated with 10 μM of Akt inhibitor (MK-2206) supplemented in the inoculum or cell culture media. At 2 hpi, cells were collected for RNA extraction and IL-1Ra qRT-PCR. IL-1Ra expression is graphed as fold-change compared to uninfected controls and normalized to actin. Tukey's One-Way ANOVA with multiple comparisons test was performed for statistical analysis. Each symbol represents the average of three technical replicates ($n = 5$–8). *$p < 0.05$, ***$p < 0.001$, ****$p < 0.0001$.

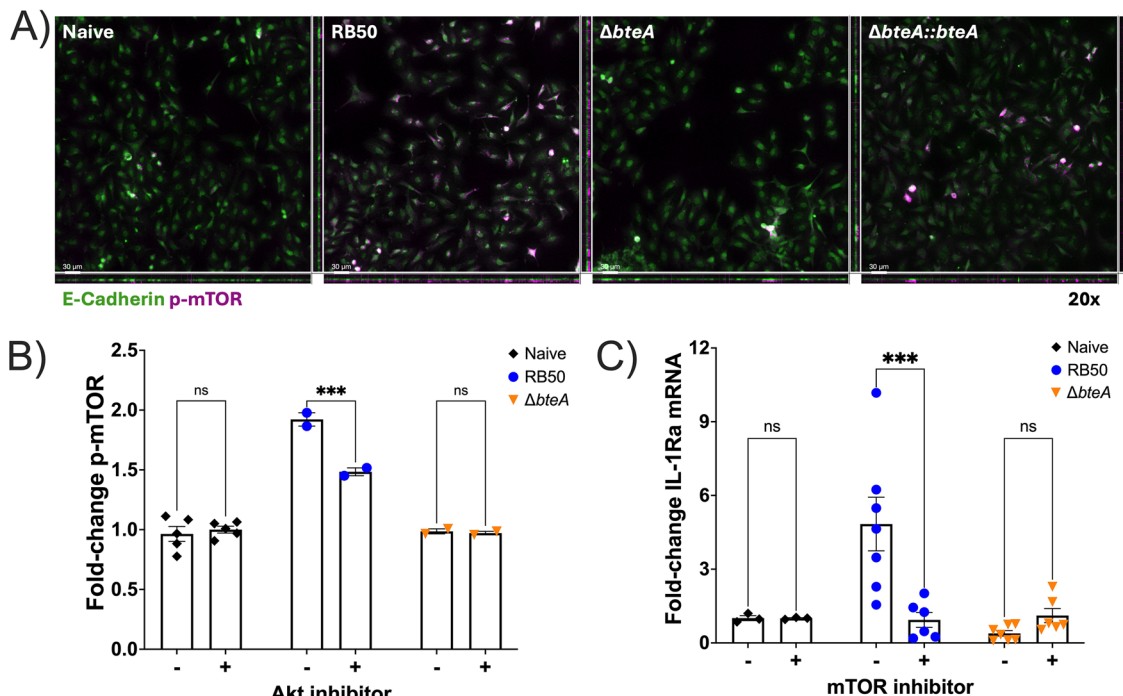

**Fig. 7 | Akt/mTOR activation is required for IL-1Ra production.** A549 cells were incubated with only cell culture media (black diamonds), or at an MOI = 10 with RB50 (blue circles) or RB50Δ*bteA* (orange inverted triangles). **A** At 2 hpi, cells were washed and fixed in 4% PFA before staining for E-cadherin (green) as a membrane marker and phospho-mTOR (purple). Images were acquired with a Cytation C-10, and a representative image of 3 biological replicates is shown, including an RB50Δ*bteA::bteA* using the same infection conditions. **B** At 2 hpi, an ELISA detecting phosphorylated mTOR (Ser2448) was performed with or without Akt inhibitor (MK-2206) treatment with the bacterial inoculum. **C** At 2 hpi following incubation with or without the mTOR inhibitor rapamycin, cells were collected for RNA extraction and IL-1Ra qRT-PCR. IL-1Ra expression is graphed as fold-change compared to uninfected controls and normalized to actin. Tukey's Two-Way ANOVA with multiple comparisons test was performed for statistical analysis. Each symbol represents the average of three technical replicates ($n = 2$-7). ns non-significant, ***$p < 0.001$.

phosphorylation of mTOR with IL-1Ra expression levels, we evaluated the fold-change of IL-1Ra with and without treatment with the mTOR inhibitor, rapamycin (10 μM). Our results show that IL-1Ra transcription levels are increased in the RB50-infected A549 cells (Fig. 7C). However, this increase in IL-1Ra expression was significantly reduced following rapamycin treatment. This mimics the levels of expression seen in RB50Δ*bteA*-infected A549 cells, with or without rapamycin treatment, indicating that BteA-mediated IL-1Ra gene transcription is dependent on mTOR

phosphorylation at S2481. Overall, our results support that activation of the Akt/mTOR host signaling pathway drives BteA-mediated IL-1Ra expression (Graphical Abstract).

## Discussion

Pathogenic bacteria must exercise optimal genome flexibility to survive in their environments within their target host, requiring fine-tuned control of virulence gene selection[48], and in our previous work, we have identified BtrS

as a key regulator of immunosuppressive pathways[13,43]. In fact, what possibly dictates the success of pathogens is their ability to use an arsenal of factors to specifically target different cells to successfully achieve infection. Herein, we provide evidence of a cell-specific strategy, during which *Bordetella* spp. utilizes the T3SS effector, BteA, to target the epithelial cells-eosinophil axis to exclusively promote early IL-1Ra induction, facilitating initial infection and persistence.

It is known that one of the main features of classical *Bordetella* spp. infections is their ability to efficiently suppress host immune responses to promote long-term infection and the subsequent long-term pathology[125]; however, the underlying mechanisms that drive this remain unclear. It is important to highlight that the mucosal tissue is particularly susceptible to infections, as it possesses high permeability allowing for air and nutrient exchange, while being highly resistant to invasion, which requires tightly regulated tolerance mechanisms to be in place. Homeostasis must be very stringently regulated by specialized immune cells such as eosinophils[55], and when dysregulation happens, it leads to chronic diseases such as eosinophilic esophagitis[126,127]. In fact, the main characteristic of eosinophilic esophagitis is the remarkable tissue remodeling and associated fibrosis, which is mostly governed by the crosstalk between eosinophils and epithelial cells, and it is mostly orchestrated by signals of the IL-1 axis, which is tightly regulated upon cross-signaling of both cell types[68]. It is possible that pathogens have been selected for their ability to exploit this communication axis to facilitate colonization and persistence.

The IL-1 axis requires intricate fine-tuning to maintain homeostasis and activate the appropriate signaling pathways in response to various stimuli during classical *Bordetella* spp. infection. While the contribution of IL-1α to pathology[63,64] and IL-1β to protective immune responses[62,66] has been investigated, the impact of other members of the family is still unclear. During *B. pertussis* infection, epithelial cells, enhance IL-1 signaling, which is critical to the development of pertussis disease-associated pathology[63]. Moreover, it has been shown that the glycopeptide known as *Bordetella* tracheal cytotoxic factor drives not only IL-1α-mediated cytotoxicity but also inducible nitric oxide synthase, which can contribute to the pathology associated with IL-1α[63,64]. Macrophage-secreted IL-1β facilitates natural killer-mediated responses required for clearance[66]. Our research has revealed that production of IL-1β by macrophages is mediated by a BtrS[43]. However, previous literature has shown that IL-1β is not modulated by the T3SS of *Bordetella* spp. and, in fact, the *Bordetella* T3SS suppresses inflammatory responses[36]. We have previously reported that the anti-inflammatory axis governed by the vasoactive intestinal peptide (VIP) and its receptor vasoactive intestinal peptide receptor 2 (VPAC2) is targeted by the *Bordetella* spp. T3SS to enhance immune suppression and promote persistence[104]. Interestingly, VIP positively regulates IL-1Ra expression[106], being IL-1Ra amongst the most potent agonist cytokines. While IL-1α has been associated with pathology, IL-1β and IL-1R have been shown to be critical for clearance of *B. pertussis* from the lungs[65], the contribution of IL-1Ra remains unclear.

IL-1Ra has been associated with increased persistence of other infections[83] and treatment with IL-1Ra analogs has been shown to increase the likelihood and burden of infections[90]. Increased levels of IL-1Ra have been observed during respiratory infections such as *Yersinia pestis*[78], *Pseudomonas aeruginosa*[77], and even and spike in IL-1Ra has been shown after priming with *Bordetella* spp.[80], leading us to hypothesize that IL-1Ra induction might be responsible for bacterial lung persistence. Indeed, our results indicate that classical *Bordetella* spp. promote early IL-1Ra induction to facilitate lung persistence. Furthermore, our findings propose the use of intranasal anti-IL-1Ra mAb as an immunotherapy to treat persistent respiratory infections, as previously suggested by Bliska et al.[78] and Gander-Bui et al.[83], and possibly for other infections such as septicemia or even antibiotic-resistant bacteria. Our results revealed that the major source of IL-1Ra at all time points investigated is eosinophils and epithelial cells. In fact, this axis is critical for tissue remodeling[128,129], further supporting the idea that this might be targeted by pathogens to facilitate persistence. We cannot disregard the possibility that a later time points, other cells will contribute to the total of IL-1Ra, such as macrophages[83], monocytes[130], or neutrophils[131], however, the overall contribution of each cell type and the temporal dynamics has yet to be investigated.

Upon immunofluorescence staining of the lungs, we observed higher IL-1Ra signal localized near the bronchiole lung areas in both murine infection models (Figs. 2A and 4C, D). This is worth noting, as we envision that upon inhalation of classical *Bordetella* spp. bacterium will be first in contact with the bronchiole area. Furthermore, deployment of T3SS machinery and subsequent injection of the BteA effector are contact-dependent processes, likely triggered by attachment to host epithelial cells. This possibly explains the restricted IL-1Ra signal observed around epithelial cells lining the lungs. It is important to discuss that *B. pertussis* BteA appears to exhibit less activity due to the insertion of an alanine residue at position 503 compared to *B. bronchiseptica*, although the exact underlying mechanism is unclear[111]. Recent work using primary human nasal epithelial cells (hNECs) cultured at the air–liquid interface as a model of early infection of the nasal cavity showed that the *B. pertussis* B1917 strain predominantly resides within the mucus layer and engages minimally with the epithelial cells[132]. However, these cells were isolated from donors who were previously vaccinated and/or infected. With this in mind, as well as previous work done by Raeven et al.[80], we anticipate that early increase in IL-1Ra induction plays an important role during primary infection, but its contribution during re-infection might be more marginal, or might occur even at earlier times, due to a stunted recall response. Moreover, we have found that IL-1Ra induction also remains conserved regardless of the genetic background or age upon infection, as seen using the *B. pertussis* model in neonate C57BL/6 J mice (Fig. 4) compared to juvenile BALB/c infection (Fig. 1). This highlights the importance of BteA-mediated IL-1Ra production in facilitating persistence following classical *Bordetella* spp. infection[111]

Using a minimalistic approach in vitro, we examined the bacterial molecular mechanism driving IL-1Ra induction. Our investigations revealed that the BteA effector of the T3SS is responsible for the IL-1Ra induction, and complementation of BteA results in restoration. Our results using A549 cells in vitro reveal that BteA promotes Akt and mTOR activation to promote IL-1Ra induction. Given the increase in IL-1Ra induction observed during other infections, we can predict that this might be a conserved strategy employed by many bacteria to manipulate host cell signaling and facilitate early events for persistence.

Our results provide a fundamental understanding of the bacterial mechanisms involved in pathogenesis and immune suppression at the molecular level. Based on the previous literature by us and others, as well as our results presented herein, we constructed our current working model. Upon infection, *Bordetella* spp. utilizes the TCT, and potentially other unknown components, to trigger IL-1α-associated pathology[63,64] by a mechanism that remains unresolved. However, this pathological effect must be balanced. Thus, we hypothesize that a BtrS-regulated mechanism independent of the T3SS promotes IL-1β and caspase-1 activation in macrophages[43], facilitating the generation of adaptive immune responses[66]. Here, we show that *Bordetella* spp. also targets the host eosinophil-epithelial cell axis to promote anti-inflammatory responses, increasing IL-1Ra expression[104]. Importantly, IL-1Ra induction has been observed in multiple disease settings, including fungal infections[79,133,134], septicemia[4,83], and even infections caused by antibiotic-resistant bacteria[135] and ESKAPE pathogens[136–138]. This work provides a target for the development of immunotherapies that can aid the treatment of patients to facilitate clearance and prevent subsequent associated pathology.

## Materials and methods
### Bacterial strains and culture conditions
*B. bronchiseptica* and *B. pertussis* strains used in this study (Table 1) were cultured using Difco Bordet-Gengou (BG) agar (BD Life Sciences, Cat. #248200) supplemented with 10% sheep defibrinated blood containing 20 μL/mL streptomycin (herein indicated as BGS media) or using LB (Fisher Scientific, Cat #BP1426-500) broth in overnight cultures as previously described[8]. *Bordetella* spp. strains were also grown on BG or BGS agar as

previously described[8]. *Bordetella pertussis* clinical isolates were grown in BHI (Sigma–Aldrich, Cat #53286-500G) or BG agar[8,47,127]. The clinical *B. pertussis* isolates used in this study (Table 1) were graciously donated to us by the Centers for Disease Control and Prevention (CDC) Division of Bacterial Disease in Atlanta, GA.

For complementation of the RB50Δ*bteA* mutant (RB50Δ*bteA::bteA*), we used the pBBRI plasmid construct[139], which encodes the *bteA* allele of the *B. bronchiseptica* RB50 strain under its native promoter. The pBBRI plasmid was introduced into RB50 via bacterial conjugation with the *Escherichia coli* strain SM10λ pir. RB50 strains harboring the pBBRI plasmid were selected on BG agar supplemented with 60 μg/mL of chloramphenicol and 100 μg/mL of cephalexin, to which classical Bordetellae are naturally resistant.

### Animal usage

For the animal experiments, including juvenile BALB/c, C57BL/6 J, and *Il1rn*$^{-/-}$ mice, they purchased them originally from Jackson Laboratories (Bar Harbor, ME) and then bred them in our facilities. Our breeding colonies were kept under the care of the employees and veterinarians of the Louisiana State University Health Sciences Center Animal Care Facility in Shreveport, LA. All animal experiments were performed in accordance with the AALAC and institutional guidelines (AUP: 20-038, 22-031, 24-036), and protocols were performed as described per IACUC approval (P20-038, P-22-031, and 24-036). Infant mouse studies were performed as described per IACUC protocol 1122017 (University of Maryland School of Medicine). For animal inoculations, mice were anesthetized with 5% isoflurane before intranasal inoculation with 30 μL solutions containing up to $1 \times 10^6$ CFU/mL of the appropriate bacterial strains (Table 1) diluted in PBS.

For the juvenile (6–8 weeks) and infant (post-natal day 7) models, C57BL/6J mice (Charles River) were used in accordance with the University of Maryland, Baltimore (UMB), Institutional Animal Care and Use Committee. Mice were inoculated via nebulizer aerosolization (Pari Vios) in an enclosed container for 20 min. Either *B. pertussis* (Bp536[97]) at an $OD_{600} = 0.1$ diluted in PBS, or PBS alone, was used as the inoculum. After inoculation, infants were returned to the sire and dam, and adults were placed in cages with similarly inoculated groups. Four- or eight-day post-inoculation with either PBS or *B. pertussis*, infants and adults alike were sacrificed by $CO_2$ asphyxiation followed by cervical dislocation. Left lung tissue was placed in 2 mL PBS for enumerating CFU, and the superior lobe was placed in 500 μL of RNAlater (Sigma–Aldrich) for RNA processing.

For IL-1Ra supplementation, 25 μg/mL of murine IL-1Ra (Sigma–Aldrich, Cat. #SRP6006) was administered via intraperitoneal injection from 1 to 14 dpi. IL-1Ra was depleted using a monoclonal antibody against IL-1Ra as described previously[93]. Briefly, mice were intranasally inoculated with up to 15 or 30 μL of 500 μg/mL mouse anti-IL-1Ra[93] (Leinco Technologies, Cat. #I-668) or Rat IgG2a isotype control (Leinco Technologies, Cat. #I-1177). As described previously, anti-IL-1Ra treatment was performed starting at day 5 post-infection[77] until 14 dpi for lung extraction and measuring bacterial burden. Mice were euthanized using $CO_2$ followed by cervical dislocation at 14 dpi. Lung bacterial burden was manually determined following a 48-h incubation at 37 °C in 5% $CO_2$ (see "Bacterial strains and culture conditions" section).

### Mouse tissue collection

Mice were sacrificed, and the lungs were dissected at 7 days following intranasal inoculation with the different bacterial strains. Lungs were collected in 2 mL reinforced tubes containing 1 mL of sterilized PBS, a protease inhibitor cocktail (Thermo Scientific, Cat. #78430), and a mixture of 0.5 mm and 1.4 mm sterile glass beads (Omni International, Cat. #UX-04728-62 and #UX-04728-56). Following homogenization and plating to determine CFU/mL as an experimental control, supernatants were frozen at −20 °C until use.

### In vitro cell culture and preparation

For A549 experiments, an MOI of 10 was used for each infection according to the cell density of 90%–95% confluency in a 24-well plate ($5 \times 10^5$ cells/well). Cell inocula were diluted in base DMEM cell culture media, with uninfected negative controls being incubated with base DMEM media only. For experiments involving Akt inhibitor treatment, cells were treated with 10 μM of the inhibitor MK-2206 (APExBIO, Cat. #A3010) at the same time as bacterial inocula were added. For experiments involving rapamycin (MedChemExpress, Cat. #HY-10219) treatment, 10 μM was added at the same time as inocula. A549 cells were incubated at 37 °C in 5% $CO_2$ with an MOI of 10 for all bacterial inocula, with or without inhibitor, for 2 h. At 2 hpi, supernatants were stored at −20 °C until use for ELISA and cells or stored at −20 °C in TRIzol for RNA extraction[45] and qRT-PCR.

To retrieve bone marrow progenitor cells, BALB/c mice were sacrificed, and the femur and tibia were extracted as previously described[140]. Briefly, after the bones were flushed, the content was suspended in RPMI media, strained, and then centrifuged. The pellet was resuspended to approximately $10^6$ cells/mL in a pre-prepared base media containing supplemented recombinant murine SCF (Peprotech, Cat. #250-03-10UG) and FLT-3L (Peprotech, Cat. #250-31L-10UG) and incubated at 37 °C for 4 days to promote cell proliferation. Base medium was supplemented with IL-5 to enhance eosinophil differentiation, and the cells were extracted for experimental use after 10 days of incubation with recombinant murine IL-5 (Peprotech, Cat. #215-15-25UG). To confirm that the culture had reached the threshold of ≥98% cell differentiation, cytospin centrifugation was used, followed by Giemsa staining. For quality control, we periodically performed flow cytometry gating for CD11b$^+$SiglecF$^{Hi}$ (RRID:SCR_024775, RRID#: SCR_024781) to quantify differentiation purity for more accurate and rigorous evaluation of bmEos purity.

### Quantitative RT-PCR and gene expression analysis

At day different days post-infection, mice were sacrificed, and lungs were collected in TRIzol reagent before homogenization and stored at −20 °C until use. Total RNA was extracted and purified in accordance with the manufacturer's protocol using the PureLink RNA Mini Kit (Invitrogen, Cat. #12183018A) with PureLink DNase treatment (Invitrogen, Cat. #12185010)[45]. RNA concentrations of the purified and concentrated samples were quantified using a Nanodrop One spectrophotometer (Thermo Fisher Scientific)[8] to determine the quality and purity of each extracted sample.

The stepwise protocol provided with the LUNA® Universal One-Step qRT-PCR Kit (New England BioLabs, Cat. #E3005) was followed for sample preparation. The qRT-PCR reactions were performed with 1 μg RNA on a Bio-Rad CFX-96 in the Louisiana State University Health Shreveport Genomics Core Facility (LSU Health Sciences Center at Shreveport, LA, USA, RRID:SCR_024775).

All primer sequences used are listed below (Table 2). At 7 dpi, lung homogenate RNA was extracted for qRT-PCR to measure changes in transcript levels. For qRT-PCR data analysis, ΔΔCq values were normalized to actin. The $2^{-\Delta\Delta Cq}$ value correlates to the fold-change in ΔΔCq values when the average ΔCq value of three technical replicates from each biological

**Table 2 | Primers used in this study for qRT-PCR**

| Gene | Direction | Sequence | Species |
|------|-----------|----------|---------|
| IL1RN | Forward | TGT TCC CAT TCT TGC ATG GC | Human |
| IL1RN | Reverse | GCA GCA TGG AGG CTG GTC AG | Human |
| IL1RN | Forward | GCT CAT TGC TGG GTA CTT ACA ATGG AAT CCT GTG GCA TCC ATG AAA C | Mouse |
| IL1RN | Reverse | CCA GAC TTG GCA CAA GAC AGGTAA AAC GCA GCT CAG TAA CAG TCC G | Mouse |
| β-Actin | Forward | TGG AAT CCT GTG GCA TCC ATG AAA C | Human |
| β-Actin | Reverse | TAA AAC GCA GCT CAG TAA CAG TCC G | Human |
| β-Actin | Forward | GGC TGT ATT CCC CTC CAT CG | Mouse |
| β-Actin | Reverse | CCA GTT GGT AAC AAT GCC ATG T | Mouse |

**Table 3 | Antibodies used for in vivo immunofluorescence (IF) microscopy staining**

| Antibody | Source | Catalog No. | RRID |
|---|---|---|---|
| Goat anti-mouse IL-1Ra (Biotinylated) | R&D Systems | BAF480 | AB_2249043 |
| Streptavidin-conjugated AF488 (Ex/Em: 495/519 nm) | Invitrogen | S11223 | NA |
| Rat anti-mouse CD326/EpCAM (PE-conjugated) | BioLegend | 118206 | AB_1134172 |
| Rabbit anti-mouse PGR2/MBP (Biotinylated) | Abexxa | Abx101775 | NA |
| Goat anti-rabbit AF647 (Ex/Em: 658/675 nm) | Invitrogen | A32733TR | AB_2866492 |
| Hoechst 33342 | BD Biosciences | 561908 | AB_2869394 |

**Table 4 | Antibodies used for in vitro immunofluorescence (IF) microscopy staining**

| Antibody | Source | Catalog No. | RRID |
|---|---|---|---|
| Rabbit Phospho-mTOR mAb | Cell Signaling | 5536S | NA |
| Goat anti-rabbit AF647 (Ex/Em: 658/675 nm) | Invitrogen | A32733TR | AB_2866492 |
| Rat anti-mouse E-cadherin | Abcam | AB11512 | AB11512-1001 |
| Donkey anti-rat AF594 (Ex/Em: 590/618 nm) | Invitrogen | A-21209 | AB_2535795 |
| Hoechst 33342 | BD Biosciences | 561908 | AB_2869394 |

replicate was normalized to actin, producing the fold-change expression relative to uninfected controls for each experimental condition.

For experiments using the severe infant disease pertussis model, RNA was isolated from tissue using TRIzol reagent (Invitrogen), and cDNA was synthesized following the manufacturer's instructions for High-Capacity cDNA Reverse Transcription Kit (Applied Biosystems). cDNA was used to perform qPCR with Maxima SYBR Green/ROX qPCR master mix (Thermo Scientific) on a Quant Studio 3 Real-Time PCR system.

Bone marrow-derived eosinophils were co-cultured in a 12-well plate and challenged at an MOI = 10. At 2 hpi, RNA was extracted in accordance with the manufacturer's protocol using the PureLink RNA Mini Kit (Invitrogen, Cat. #12183018A). Briefly, following centrifugation, cells were resuspended in TRIzol reagent and treated with PureLink DNase (Invitrogen, Cat. #12185010) treatment[45]. RNA concentrations were quantified using a Nanodrop One spectrophotometer (Thermo Fisher)[8] and stored at −20 °C for qRT-PCR. For ELISA, cell supernatants were stored at −20 °C until use. Please note that the IL1RN gene exclusively encodes IL-1Ra.

**Protein quantification (ELISA)**
ELISA kits were used for quantifying protein levels of IL-1Ra (Invitrogen, Cat. #EMIL1RN), IL-1α (Invitrogen, Cat. #BMS243-2), and IL-1β (Invitrogen, Cat. #KHC0011), all performed according to the manufacturer's instructions. From each generated standard curve, unknown sample concentrations were determined by curve interpolation analysis using GraphPad Prism software. Appropriate dilution factors were multiplied to unknown concentrations as needed to calculate the final protein concentrations of each sample (pg/mL). For determining phosphorylation levels of Akt (Ser473) and mTOR (Ser2448), A549 cell culture lysates at 2 hpi were collected for phospho-Akt ELISA (Invitrogen, Cat. #KHO0111) or phospho-mTOR ELISA (Antibodies Inc., Cat. #A102225), both performed following the manufacturer's recommendations.

**Immunofluorescence (IF) staining and microscopy**
For all tissue immunostaining experiments, mice were intratracheally perfused with sterile PBS followed by 4% paraformaldehyde (PFA). Lungs were subsequently fixed overnight in 4% PFA prior to processing. For tissue processing, the lungs were paraffin-embedded and sectioned (0.5 μm), then placed on glass slides (RRID:SCR_024776). Specimens were prepared for sequential staining through deparaffinization in consecutive xylene washes, followed by

rehydration in decreasing concentrations of ethanol ranging from 100% to 50%, antigen retrieval, and blocking steps[45]. The mounted tissue samples were incubated with the according primary and secondary antibody (Table 3), following previously published methods[141]. Immunofluorescence images of the stained lung specimens were captured using an Olympus CSU W1 Spinning Disk Confocal Microscopy System (RRID:SCR_024775). Image analysis for quantification of fluorescence signal was conducted through the Imaris 3D Analysis Software (v10) (RRID:SCR_024775).

Three-dimensional analysis of IL-1Ra$^+$ eosinophils and epithelial cells was conducted using Imaris 10.1 software (Bitplane, Oxford Instruments). Nuclei and cell markers were rendered as 3D surfaces, while IL-1Ra signals, due to their punctate appearance, were rendered as 3D spots. Nuclei were defined using DAPI staining and rendered as 3D surfaces. Eosinophils were identified based on MBP (major basic protein) staining, and epithelial cells were identified using EpCAM (epithelial cell adhesion molecule) staining. Both cell markers were rendered as 3D surfaces using the "Surface" tool in Imaris. IL-1Ra signals were rendered as 3D spots using the "Spots" tool, with the spot diameter set according to the average size of the punctate IL-1Ra signal observed in the images. Spot creation parameters were optimized to ensure an accurate representation of the signal. Following object rendering, classification was performed using the "Filter" tool in Imaris to segregate objects based on specific criteria. Eosinophils were defined as MBP$^+$ surfaces located within 0.05 μm of the nucleus surface. Eosinophils were classified as IL-1Ra$^+$ if a spot was located within 0.05 μm of the eosinophil surface. For the analysis of epithelial cells, regions of interest containing visible bronchial epithelium were selected. Epithelial cells were defined as EpCAM$^+$ surfaces within 0.05 μm of the nuclear surface. Epithelial cells were classified as IL-1Ra$^+$ if a spot was located within 0.05 μm of the epithelial cell surface. The number of cells meeting each classification criterion (e.g., IL-1Ra$^+$ eosinophils, IL-1Ra$^+$ epithelial cells) was quantified for each image. The analysis was repeated for at least three independent images to ensure reproducibility. The results were averaged and reported as the final quantification of IL-1Ra$^+$ eosinophils and epithelial cells. Statistical analysis and plotting of these results were done using GraphPad Prism (v10.0.2).

For in vitro cell staining, A549 cells were washed with PBS at 2 hpi following incubation steps, fixed with 4% PFA for 30 min. For immunofluorescence (IF) staining and microscopic imaging, A549 cells were incubated with primary and secondary antibodies (Table 4). We conducted a three-dimensional analysis of phospho-mTOR intensity in A549 cells using Imaris v10.2 (Bitplane, Oxford Instruments). To prepare images for precise 3D rendering, each channel was processed with background subtraction and deconvolution filters, reducing noise and enhancing clarity in the raw image. The Cy5 channel, which contained the phospho-mTOR stain, was reconstructed as a 3D surface. To improve analysis accuracy, we manually removed any surfaces that appeared non-specific or likely false positives. The mean intensity of each mTOR surface was then calculated using the Imaris Vantage Plot feature and displayed in a 1D plot. These intensity values for each group were exported to GraphPad Prism (v10.0.2) for further visualization and statistical analysis.

**Institutional core facilities**
LSU Health Shreveport Research Core Facilities (RRID:SCR_024775) were used for performing FACS analysis (Flow Cytometry Core), qRT-PCR

(Genomics Core), as well as Imaris usage and analysis for microscopy images (Microscopy Core). LSU Health Shreveport Center for Redox Biology and cardiovascular diseases was used for the embedding, sectioning, and pathology (RRID:SCR 024776).

## Statistics and reproducibility

All experiments were performed using at least 2–3 independent biological replicates[142]. The exact number of mice and technical replicates is indicated in each figure legend. All datasets being graphed and analyzed for statistical significance had outliers identified and removed using the ROUT method parameters ($Q = 1\%$). Unless stated otherwise, Dunnett's Two-way ANOVA with multiple comparisons was used for statistical significance analyses of grouped experiments containing multiple variables (e.g., infection settings with treated and untreated groups), or Tukey's One-way ANOVA with multiple comparisons for statistical analyses of datasets graphed in individual columns (e.g., infections comparing groups at a single time point post-inoculation). All statistical analysis was done using GraphPad v10.0. A $p$-value $\leq 0.05$ was considered as statistically significant. In all graphs included in the figure panels, asterisks indicate the following values of statistical significance: $* = \leq 0.05$, $** = \leq 0.01$, $*** = \leq 0.001$, and $**** = \leq 0.0001$.

## Reporting summary

Further information on research design is available in the Nature Portfolio Reporting Summary linked to this article.

## Data availability

All raw data files have been published to a public Figshare repository (https://doi.org/10.6084/m9.figshare.29832140). Raw microscopy images or additional data files are available upon request to the corresponding author of this article.

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

## Acknowledgements

We would like to acknowledge the support of the following funding bodies: NIH COBRE award (NIGMS P20-GM134974); "Center of Excellence for Arthritis and Rheumatology intramural award"; Intramural Research Council Seed support package from LSU Health Sciences Center at Shreveport; and start-up package from LSU Health Sciences Center at Shreveport. The authors would like to acknowledge the CAIPP Bioinformatics and Modeling core (RRID:SCR_024871) and the CAIPP Immunophenotyping core (RRID:SCR_024781) from the COBRE grant "Center for applied immunological and pathological processes" of NIGMS (P20-GM-139474). The authors acknowledge the Animal Models & Histology core of the COBRE Center for Redox Biology and Cardiovascular Disease (NIGMS P20 GM121307) for the histology and pathology support. We would like to acknowledge the "Multidisciplinary Training in Cardiovascular Pathophysiology Fellowship" (NIH-NHLBI #5T32HL155022-03) pre-doctoral T32 fellowship that provides financial support to K.M.P. We would like to acknowledge the Department of Microbiology and Immunology for their support, provided by the Ken Peterson Memorial Predoctoral Fellowship granted to K.M.P. We would also like to acknowledge the Ike Muslow Predoctoral Fellowship that provided financial support to N.F. The authors would also like to acknowledge the SMART program, funded by the Biomedical Research Foundation of NW Louisiana, that supported S.J., while she was in our laboratory in her senior year of High School. J.K. was supported by the grants 21-05466S of the Czech Science Foundation (www.gacr.cz) and Talking microbes - understanding microbial interactions within One Health framework (CZ.02.01.01/00/22_008/0004597) of the Ministry of Education, Youth and Sports of the Czech Republic (www.msmt.cz). K.M.S. and J.K. were supported by NIH-NIAID R21 AI163595. We would like to acknowledge Dr. Martin Sapp, Dr. Matthew Woolard, Dr. Rona Scott, and other members of the Microbiology and Immunology Department at LSU Health Science Center at Shreveport for the brainstorming and support during the development of this project. We would like to thank Lucia Tondella and Michael Weigand, from the Centers for Disease Control and Prevention, for the selection and contribution of classical *Bordetella* spp. clinical strains. We would also like to acknowledge the inspirational brainstorming sessions with Dr. Jonathan Kagan. We would also like to acknowledge Laura Lopez-Candales for her contribution to experiments performed during the summer through the Fundacion Barrie de la Maza and the LSU Health Science Center Undergraduate Research Program. The Graphical Abstract was created using BioRender (https://BioRender.com/y35icia).

## Author contributions

K.M.P. performed in vitro and in vivo experiments, analyzed data, contributed intellectually to the project, and to writing and editing. N.F. performed lung staining, analyzed data, and contributed to writing and editing. J.K. contributed with several strains, brainstorming, editing, and contributed with editing. T.L/W. performed in vitro experiments, analyzed data, and contributed to editing. S.J. performed in vitro experiments, analyzed data, and contributed to editing. J.K. performed the juvenile C57BL6/J experiments. K.M.S. performed the juvenile and neonate C57BL6/J experiments and contributed to editing. N.P.A., S.M., and C.S. contributed to brainstorming and editing. M.C.G. conceptualized the project, designed and performed the experiments, analyzed data, wrote and edited the paper, obtained funding, and supervised.

## Competing interests

Monica Cartelle Gestal submitted the patent titled: "Anti-IL-1RA as treatment for long-term bacterial infection" through LSU Health Sciences Center at Shreveport (PCT #2932719-000237-WO1; LSUHSC-S-2022-010-03). All other authors declare no competing interests for this paper.
