## [Transparent Peer Review file · Communications Biology]

The Bordetella type III secretion system effector BteA targets host eosinophil-epithelial cell signaling to promote IL-1Ra expression and persistence.

Corresponding Author: Dr Monica Gestal

Version 0:

Reviewer comments:

Reviewer #1

(Remarks to the Author)

This study demonstrates the role of the type III secretion effector BteA of *Bordetella pertussis* and *Bordetella bronchiseptica* in the modulation of eosinophil signalling and the production of IL-1Ra. Immune modulation by *B. pertussis* and *B. bronchiseptica* is both complex and vital for persistence of the pathogen in the host. This study demonstrates that BteA is vital in allowing the pathogen to persist and that this action is mediated via eosinophils. This study furthers our understanding of *Bordetella* modulation of host immune responses that is of broad interest.

Experimental approach is well explained, and the authors do well not to over state their findings but carefully support their conclusions with the results. The statistical analysis has been done with sufficient rigor and explained well.

Major:

41,47, 48, etc 404: You state that this is a strategy used *Bordetella* spp. but you have only presented data from *Bordetella pertussis* and *Bordetella bronchiseptica*, the references you use only refer to *B. pertussis*, *B. bronchiseptica* and *Bordetella parapertussis*. However, the genus *Bordetella* includes several non-pathogenic species, do they possess BteA, and is its function conserved? If BteA is only present in the classical *Bordetella*, I suggest you revise your wording to reflect that this is a strategy specific to those species. As currently written, I am concerned that the claim overextends to species for which no evidence is provided.

Minor:

102: 'ideal targets by pathogens' should read 'of pathogens'

Clostridioides difficile as this is the first mention.

330/335: inconsistent use of 'wildtype/wild-type'

340: sentence appears incomplete 'where at 14 dpi'

Reviewer #2

(Remarks to the Author)

The author demonstrated that IL-1Ra responses mediated by eosinophils and epithelial cells are important for clearing *B. pertussis* bacteria from the lungs. This response is crucial and present as early as in neonates. The induction of IL1Ra depends on a functional T3SS, and more specifically on the presence of the BteA effector. The BteA-mediated induction of the IL1Ra response occurs through increased Akt/mTOR phosphorylations, leading to IL1Ra expression. The manuscript is very well written, clear and easy to follow.

Here are a few of my remarks.

On lines 112–113, the authors described the role of IL-1Ra binding as 'promoting anti-inflammatory responses'. However, it merely blocks the binding of IL-1 α and IL-1 β , so it does not have anti-inflammatory properties, but rather blocks the IL-1R-mediated pro-inflammatory response. Does IL-1Ra binding itself have an active anti-inflammatory response through binding to IL-1R? Can you rephrase 'promoting' or 'yielding to'?

On Fig. S2, I would remove the title 'B. pertussis RNA-seq' because the RNA-seq is of infected mice, not the bacteria. I would also change the axis labelled 'Il1rn gene normalized counts'.

Line 220 it is Fig. S2B not Fig. S2A

Line 249: I don't think you can use human symptomatology to draw conclusions for the mouse challenge experiment to determine the optimal day for anti-IL1Ra mAb treatment. The results presented in Fig S1B are sufficient.

For lanes 264–269, what is the rationale for selecting these BP clinical isolates? Are they PtxP1 or PtxP3; PRN-proficient? Do they produce Fim2 or Fim3?

In lane 279, it is written that IL-1Ra expression increases as early as four days post-infection, while in Fig. S1C, it is three days post-infection, and in Fig. 4A, it is four days post-infection.

Lane 313 it is Fig. S2A not Fig. S2B

Lane 333 it is fig 5D not 5E

Can you rephrase the line 340 there is a missing word "then were sacrificed at 14 dpi ? "

Lane 384 it is Fig7C not 7B ; it is not a decrease of mTOR phosphorylation but a limitation of the mTOR phosphorylation

Lane 385 addition

Lane 394 the word abolished is too strong maybe reduced

Could you rephrase line 364? As it is written, it suggests that BteA is the only way to mediate Akt phosphorylation. Could you perhaps say something like, 'Our results indicate that Akt phosphorylation at the S473 site is increased in the presence of bacteria that produce BteA', or 'Our results indicate that BteA induces Akt phosphorylation at the S473 site'?

The persistence of Bordetella in the lung, mediated by BteA-dependent induction of IL-1Ra, must be released to enhance the epithelial cell-eosinophil axis and block host defences involved in bacterial clearance. Does co-infection with a mixture of BteA-depleted and BteA-proficient strains increases the persistence of the BteA-depleted strain in the lung? The same issue applies to the usage of a BopN-depleted strain, which has been shown to increase BteA secretion and may also increase the IL-1Ra response, thereby enhancing the persistence of a BteA-depleted strain in the lung (see reference 112 in the manuscript). This proposed mixed animal infection study may strengthen the results of the manuscript.

Version 1:

Reviewer comments:

Reviewer #2

(Remarks to the Author)

The manuscript is now ready for publication.

REVIEWERS COMMENTS:

Reviewer #1 (Remarks to the Author):

This study demonstrates the role of the type III secretion effector BteA of *Bordetella pertussis* and *Bordetella bronchiseptica* in the modulation of eosinophil signalling and the production of IL-1Ra. Immune modulation by *B. pertussis* and *B. bronchiseptica* is both complex and vital for persistence of the pathogen in the host. This study demonstrates that BteA is vital in allowing the pathogen to persist and that this action is mediated via eosinophils. This study furthers our understanding of *Bordetella* modulation of host immune responses that is of broad interest.

Experimental approach is well explained, and the authors do well not to over state their findings but carefully support their conclusions with the results. The statistical analysis has been done with sufficient rigor and explained well.

We would like to thank the reviewer for the suggestions and the edits suggested that we believe have significantly improve the manuscript. Thanks for the reviews and the positive words above stated.

Major:

41,47, 48, etc 404: You state that this is a strategy used *Bordetella* spp. but you have only presented data from *Bordetella pertussis* and *Bordetella bronchiseptica*, the references you use only refer to *B. pertussis*, *B. bronchiseptica* and *Bordetella parapertussis*. However, the genus *Bordetella* includes several non-pathogenic species, do they possess BteA , and is its function conserved? If BteA is only present in the classical *Bordetella*, I suggest you revise your wording to reflect that this is a strategy specific to those species. As currently written, I am concerned that the claim overextends to species for which no evidence is provided. We would like to thank the reviewer for pointing this very critical aspect to us. This is totally correct, we should have been more specific with the *Bordetella* spp. that we are discussing in this manuscript, which only includes classical *Bordetella* spp. We have now revised the paper with added specifications of classical *Bordetella* species.

Minor:

102: 'ideal targets by pathogens' should read 'of pathogens' We would like to thank the reviewer for the correction. We have now replaced "by" for "of".

Clostridioides difficile as this is the first mention. Thank you for picking up on this, we have now spelled out "*Clostridioides difficile*".

330/335: inconsistent use of 'wildtype/wild-type' Thank you for this comment and we have now used "wildtype" consistently across the manuscript.

340: sentence appears incomplete 'where at 14 dpi'. We would like to thank the reviewer for this observation, and we have now corrected this sentence as well as the previous to now achieve a compelling message.

Reviewer #2 (Remarks to the Author):

The author demonstrated that IL-1Ra responses mediated by eosinophils and epithelial cells are important for clearing *B. pertussis* bacteria from the lungs. This response is crucial and present as early as in neonates. The induction of IL1Ra depends on a functional T3SS, and more specifically on the presence of the BteA effector. The BteA-mediated induction of the IL1Ra response occurs through increased Akt/mTOR phosphorylations, leading to IL1Ra expression. The manuscript is very well written, clear and easy to follow.

We would like to thank the reviewer for the encouragement and positive words about our manuscript. The suggestions are very positive, and we believe these changes and edits have positively impact the quality of our manuscript.

Here are a few of my remarks.

On lines 112–113, the authors described the role of IL-1Ra binding as 'promoting anti-inflammatory responses'. However, it merely blocks the binding of IL-1 α and IL-1 β , so it does not have anti-inflammatory properties, but rather blocks the IL-1R-mediated pro-inflammatory response. Does IL-1Ra binding itself have an active anti-inflammatory response through binding to IL-1R? Can you rephrase 'promoting' or 'yielding to'? We would like to thank the reviewer for this particular comment. We have now been more specific with our language and revised not only those two lines, but the whole manuscript to clarify that IL-1Ra suppress pro-inflammatory responses, and that it is an antagonist, rather than

calling it “anti-inflammatory”. This is a very good point, and this clarification has strengthened our message.

On Fig. S2, I would remove the title '*B. pertussis* RNA-seq' because the RNA-seq is of infected mice, not the bacteria. I would also change the axis labelled 'Il1rn gene normalized counts'. We have now removed the label of the axis as well as edit the name to clearly state that the sequencing corresponds with murine lungs (please see updated Supplementary Figure 1C).

Line 220 it is Fig. S2B not Fig. S2A Thank you for realizing this labeling error. This is now corrected.

Line 249: I don't think you can use human symptomatology to draw conclusions for the mouse challenge experiment to determine the optimal day for anti-IL1Ra mAb treatment. The results presented in Fig S1B are sufficient. We have removed the references to humans.

For lanes 264–269, what is the rationale for selecting these BP clinical isolates? Are they PtxP1 or PtxP3; PRN-proficient? Do they produce Fim2 or Fim3? Thank you for bringing this up to our attention as it is very important information. We have now added the characteristic of these strains as well as the references where the clinical isolates were sourced.

In lane 279, it is written that IL-1Ra expression increases as early as four days post-infection, while in Fig. S1C, it is three days post-infection, and in Fig. 4A, it is four days post-infection. We acknowledge that this was confusing, we have done the RNA at day 3 post-infection but based on the timeline and life expectancy of these mice (dying by day 9), we performed the following experiments at day 4 (which will be halfway through the infection process).

Lane 313 it is Fig. S2A not Fig. S2B Thank you for realizing of the labeling error, this is now corrected.

Lane 333 it is fig 5D not 5E Thank you for realizing of the labeling error, this is now corrected.

Can you rephrase the line 340 there is a missing word “then were sacrificed at 14 dpi ? “ We have changed this sentence, as well as the previous.

Lane 384 it is Fig7C not 7B ; it is not a decrease of mTOR phosphorylation but a limitation of the mTOR phosphorylation Thank you for bringing this to our attention. This is now corrected, and this entire section has been edited for clarity.

Lane 385 addition Thank you for catching this error. We have corrected the spelling.

Lane 394 the word abolished is too strong maybe reduced We replaced “abolished” with “reduced” here.

Could you rephrase line 364? As it is written, it suggests that BteA is the only way to mediate Akt phosphorylation. Could you perhaps say something like, 'Our results indicate that Akt phosphorylation at the S473 site is increased in the presence of bacteria that produce BteA', or 'Our results indicate that BteA induces Akt phosphorylation at the S473 site'? We have replaced this sentence with the one suggested here by the reviewer.

The persistence of Bordetella in the lung, mediated by BteA-dependent induction of IL-1Ra, must be released to enhance the epithelial cell-eosinophil axis and block host defences involved in bacterial clearance. Does co-infection with a mixture of BteA-depleted and BteA-proficient strains increases the persistence of the BteA-depleted strain in the lung? The same issue applies to the usage of a BopN-depleted strain, which has been shown to increase BteA secretion and may also increase the IL-1Ra response, thereby enhancing the persistence of a BteA-depleted strain in the lung (see reference 112 in the manuscript).

This proposed mixed animal infection study may strengthen the results of the manuscript. We would like to deeply thank the reviewer for this suggestion. While it is not included in this manuscript, this experiment is planned for our future research to deepen the mechanistic understanding of BteA-BopN crosstalk in host immune suppression.